**Subject Category:**
Biology (whole organism)

behaviour/cognition/ecology

social learning, fear learning, cognition, human–wildlife conflict, urban ecology, corvid

**Authors for correspondence:**
Victoria E. Lee
e-mail: vel202@exeter.ac.uk
Alex Thornton
e-mail: a.thornton@exeter.ac.uk

# Social learning about dangerous people by wild jackdaws

Victoria E. Lee[1], Noémie Régli[2], Guillam E. McIvor[1] and Alex Thornton[1]

[1]College of Life and Environmental Sciences, University of Exeter, Penryn Campus, Penryn, Cornwall TR10 9FE, UK
[2]Faculté des Sciences et Techniques, Université Jean Monnet, 23 Rue du Dr Paul Michelon, 42100 Saint-Étienne, France

(iD) VEL, 0000-0003-3981-387X

For animals that live alongside humans, people can present both an opportunity and a threat. Previous studies have shown that several species can learn to discriminate between individual people and assess risk based on prior experience. To avoid potentially costly encounters, it may also pay individuals to learn about dangerous people based on information from others. Social learning about anthropogenic threats is likely to be beneficial in habitats dominated by human activity, but experimental evidence is limited. Here, we tested whether wild jackdaws (*Corvus monedula*) use social learning to recognize dangerous people. Using a within-subjects design, we presented breeding jackdaws with an unfamiliar person near their nest, combined with conspecific alarm calls. Subjects that heard alarm calls showed a heightened fear response in subsequent encounters with the person compared to a control group, reducing their latency to return to the nest. This study provides important evidence that animals use social learning to assess the level of risk posed by individual humans.

## 1. Introduction

In changing environments, social information allows organisms to learn about novel threats without the need for potentially costly encounters [1]. Despite having received less attention than in foraging contexts [2], the importance of social learning in avoiding predators has been demonstrated in a range of taxa (insects [3], fishes [4,5], birds [6–8], mammals [9–11]; see [2] for a review). However, with a few exceptions [12,13], most studies investigating aspects of socially acquired predator avoidance have been conducted in captivity. Field studies carried out under natural conditions are urgently needed in order to establish how social information influences antipredator behaviour in real-world

settings [2,14,15]. As socially-acquired predator avoidance is hypothesized to confer benefits when predation risk varies in space and time [16], when new predators are encountered [17] or when community composition is altered [2], understanding how social environments shape antipredator responses is vital in predicting and mitigating the effects of environmental change [18]. Furthermore, if potential predators vary in their level of threat, the ability to discriminate between individual predators of the same species is likely to be beneficial [12]. In this scenario, social learning may help prey to fine-tune their antipredator behaviour to avoid the costs of fleeing in response to benign encounters.

For many animals today, humans present a threat greater than or equivalent to natural predators [16,19–21], but individual people can vary substantially in their behaviour towards wildlife [12,22]. In urban or agricultural areas, species that exploit the opportunities and resources provided by human activity are often persecuted [8,12], but not all people will be involved in persecution. In these cases, fear of humans can allow individuals to avoid danger, but exhibiting an inappropriate fear response to all humans would prevent individuals from exploiting these habitats. The ability to learn to discriminate between threatening and non-threatening people may therefore be highly beneficial. For example, several urban bird species have demonstrated the ability to learn about dangerous people based on previous experience, and even incorporate subtle human cues such as gaze direction when assessing risk (northern mockingbird *Mimus polyglottos* [22], Eurasian jackdaw *Corvus monedula* [23,24], American crow *Corvus brachyrhynchos* [25,26], raven *Corvus corax* [27], Eurasian magpie *Pica pica* [28], green bee-eater *Merops orientalis* [29], European starling *Sturnus vulgaris* [30]). In this context, learning from the responses of conspecifics could be extremely valuable in allowing individuals to learn which humans are dangerous without the risk of a potentially fatal encounter. However, the potential for animals to use social learning to discriminate between dangerous and harmless people is poorly understood. Determining how social learning shapes responses to humans may provide an additional insight into how species and populations cope in a rapidly changing anthropogenic landscape [18,31].

A small number of studies raise the possibility that social information plays a key role in shaping how individuals learn to associate people with a dangerous place or event. For example, common myna (*Acridotheres tristis*) become warier in a location where they have previously seen an alarmed bird being captured by a human, relative to a control treatment where subjects were not given conspecific cues [8,32]. A field study of American crows (*C. brachyrhynchos*) provides the most compelling evidence to date that information about dangerous humans is socially transmitted through populations [12]. Here, experimenters wearing masks when trapping crows were later mobbed by birds that were not present at the initial capture event, and offspring present during mobbing events involving parents later mobbed masks independently. This evidence is indicative of both horizontal and vertical transmission of social information, but the processes through which social learning might occur remain unclear as individuals' exposure to social information was not manipulated explicitly. Mobbing events, during which large groups of crows gathered and alarm-called for extended periods (up to 40 crows for up to 10 min) are likely to have created significant opportunities for social learning, but this possibility has yet to be tested experimentally. Moreover, for many animals such large-scale mobbing of dangerous people is likely to be infrequent because of the substantial costs (in terms of time, energy and risk) [33] associated with approaching a potential threat. More commonly, individuals will be exposed to a short-lived bout of conspecific alarm calling, often from a single individual who may be out of sight. If individuals are able to learn from acoustic cues alone, this could provide a powerful mechanism by which information about dangerous people could spread through populations without the need for involvement in costly mobbing events. We therefore tested whether exposure to alarm calls from a single conspecific is sufficient to change the responses of naive wild birds towards unfamiliar humans.

We also examined whether learning was influenced by the characteristics of the signaller. Learning indiscriminately from others may be detrimental if information is irrelevant, erroneous or out of date [34,35], so individuals are expected to employ social learning strategies in deciding when, how and from whom to learn [1,36]. However, no studies have empirically investigated the social learning strategies employed in antipredator contexts (but see [10,13,37,38]). This is surprising given that social dynamics may play a key role in shaping predator recognition: for example, when novel predators are encountered frequently or individual predators differ in their level of threat, prey may preferentially attend to information from familiar conspecifics who possess more locally relevant knowledge about danger [1,36]. To investigate this possibility, we used alarm calls from familiar and unfamiliar conspecifics to determine whether the familiarity of the caller influences subsequent responses to a potential threat.

We conducted experiments on wild jackdaws (*C. monedula*), highly social members of the corvid family that form long-term monogamous pair bonds and tend to nest in breeding colonies that are stable over time [39,40]. They are commonly found in agricultural and urban areas across Europe, exploit anthropogenic resources (including food and breeding sites) [41,42] and are targeted as pests [43,44]. Like many corvids [25–28], jackdaws can learn to discriminate between individual people based on facial cues [23,24]. When presented with a person wearing a mask, jackdaws return to their nest more quickly if that mask has previously been worn during a nest intrusion, demonstrating that jackdaws learn to recognize individual people and associate them with prior events [24]. In order to test whether jackdaws use social learning during encounters with people, we presented breeding jackdaws with an unfamiliar person at their nest-box, using playback of alarm calls from colony and non-colony members to provide social information about danger. If birds incorporated the social information from the playback into their behavioural response, we predicted that birds that heard alarm calls would subsequently show a higher fear response relative to a control group that heard playback of contact calls. As alarm calls from familiar conspecifics have previously been shown to increase collective responses to threats in jackdaws [45], we also predicted heightened fear responses among individuals that had heard the alarm calls of colony members rather than unfamiliar birds. Using a three-phase, within-subject paradigm previously employed in studies of socially acquired predator avoidance [8,46], we were able to quantify changes in individual response according to playback treatment, without altering the nature of the encounter with the human. Thus, we were able to separate social learning from the effects of individual learning and ensure that the nature of the social learning experience was consistent between test subjects.

## 2. Methods

### 2.1. Study population

This experiment was conducted during the 2017 breeding season using free-living nest-box populations of jackdaws at three study sites in Cornwall, UK: a village churchyard (Stithians 50°11′26″ N, 5°10′51″ W; 33 nest-boxes), an active farmyard (Pencoose Farm 50°11′56″ N, 5°10′9″ W; 35 nest-boxes), and at the University of Exeter's Penryn campus (50°17′32″ N; 5°11′96″ W; 11 nest-boxes). At these sites, jackdaws are captured in nest-boxes or ladder traps and individually colour-ringed by the Cornish Jackdaw Project. Resident jackdaws at all three sites have experience of humans walking around the area on a regular basis without posing any threat. However, some persecution of corvids occurs in the area (V. E. Lee and G. E. McIvor 2016-2017, personal observation) and fieldworkers monitor nests daily during the breeding season, eliciting alarm calling from resident jackdaws. Thus, discriminating between dangerous and harmless people is likely to be beneficial.

### 2.2. Experimental design

Following protocols used in previous studies of socially acquired predator avoidance [8,46], focal jackdaws underwent three trial phases (figure 1):

  (i) baseline phase: subjects presented with a novel human stimulus (an experimenter wearing a mask) at their nest-box;
 (ii) training phase: subjects presented with the same human stimulus, paired with playback of conspecific calls to provide social information about the level of danger (see below); and
(iii) test phase: subjects presented with the human stimulus a third time, to compare any changes in individual behaviour occurring as a result of the training.

In the training phase, scold calls were used to imply danger and contact calls were used as a control (electronic supplementary material, figure S1). Scold calls are antipredator vocalizations given by jackdaws to recruit others to mob a predator [45], and contact calls are used in a range of contexts to advertise identity but are not associated with any specific event [46]. Contact calls and scold calls both encode information about the identity of the caller [45,47], and are frequently heard in jackdaw colonies. To determine whether familiarity with the caller influences social learning, calls were presented from colony members and unfamiliar birds from a different breeding colony. Focal pairs

| treatment | | presentation (i) (baseline) | presentation (ii) (training) | presentation (iii) (test) |
|---|---|---|---|---|
| scold calls | familiar | = | 🔊 | ‼️ |
| | unfamiliar | = | 🔊 | ❗ |
| contact calls | familiar | = | 🔊 | = |
| | unfamiliar | = | 🔊 | = |

**Figure 1.** Experimental design and predictions. Focal birds received three stimulus presentations at their nest-box (baseline, training and test phase). In the training phase, subjects were presented with playbacks from one of four treatments (scold call/familiar; scold call/unfamiliar; contact call/familiar; contact call/unfamiliar). If jackdaws learn socially about dangerous people, we expected birds to increase their fear response to the human following association with scold calls (denoted by exclamation marks). If jackdaws engage in directed social learning in this context, we predicted that the strength of the effect would be greater for birds that heard familiar scold calls in the training (denoted by two exclamation marks).

were assigned to one of four treatments: familiar contact calls, unfamiliar contact calls, familiar scold calls or unfamiliar scold calls. If subjects incorporate social information from the training phase into their behavioural response, we expected an increase in fear behaviour in the test phase compared to the baseline phase, but only for birds in the scold call treatments (figure 1). If subjects preferentially attend to social information from familiar conspecifics, subjects that heard familiar scold calls in the training phase were predicted to show a greater change in fear response between the baseline and test phases, compared to subjects that heard unfamiliar scold calls.

## 2.3. Experimental stimuli: presentation of unfamiliar human

Experimenters wore full-head latex masks throughout all trials to ensure the novelty of the stimulus and avoid potential confounding effects of birds' familiarity with the experimenters (electronic supplementary material, figure S2). Two different masks were worn at each site, one during scold call trials and the other during contact call trials. The mask-treatment combinations were counterbalanced between sites; the mask worn during scold call trials at one site was worn during contact call trials at another site and vice versa. As commercially available masks have lurid, unrealistic hair or lack hair, all masks were paired with a plain hat. Mask and hat pairings were kept constant at each site.

Two experimenters (V.E.L. and N.R.) carried out trials, with focal birds being presented with the same experimenter for all three trials. Both experimenters carried out trials in all treatment groups at all three sites, and wore both types of mask. Experimenters wore a large raincoat to disguise any body shape or gait cues and wore the same clothing during all trials.

## 2.4. Experimental stimuli: playback calls

### 2.4.1. Audio recordings

Playback calls were extracted from recordings obtained during the 2014–2016 breeding seasons, and only calls of known individuals were used in the experiment. Contact calls were recorded using lapel microphones (AKG-C417PP) installed inside nest-boxes, and scold calls were recorded using a shotgun microphone (Sennheiser ME66) while nests were being visited by researchers. All calls were recorded using multitrack linear PCM recorders (Olympus LS-100 & Tascam DR-100MKII).

### 2.4.2. Call extraction

Exemplars of contact calls and scold calls with minimal background noise were extracted from audio recordings and normalized for amplitude using AUDACITY (www.audacityteam.org). Extracted calls were arranged into playback files comprising either five contact calls or five scold calls from a single individual, occurring at 2 s intervals to simulate natural calling (electronic supplementary material, figure S1). Where possible, five unique calls were used in playbacks (contact calls: 3–11 calls from 30 individuals, mean 5 calls per individual; scold calls: 3–13 calls from 18 individuals, mean 4 calls per individual; see the electronic supplementary material for details). All playbacks were played through FoxPro Fury remote-controlled loudspeakers, at a set volume level that simulated the natural amplitude of calls at a distance of 10 m [45,46].

### 2.4.3. Allocation of playbacks

Focal pairs were assigned to treatment groups as required to maintain a balanced design across the experiment. For 'familiar' treatment playbacks, near neighbours (birds nesting within 500 m of the focal pair) were used wherever possible to maximize the likelihood that focal birds were familiar with the caller. However, we avoided using the calls of immediate neighbours (birds nesting within 200 m of the focal pair), who were likely to have been in the vicinity during trials. For 'unfamiliar' treatment playbacks, focal pairs were played calls from non-colony members. Because of their close geographical proximity (1.2 km) and the observed movement of birds between Stithians churchyard and Pencoose farm (V. E. Lee and G. E. McIvor 2016-2017, personal observation), only calls obtained from birds at the Campus site (more than 5 km) were used in the unfamiliar treatments at the Stithians and Pencoose sites (extensive observations suggest that birds do not move between Stithians/Pencoose and Campus sites). Both male and female callers were used for playbacks, allocated as required to balance the proportion of callers of both sexes within sites and across treatment groups. All callers were known to be alive and breeding in the colonies at the time of the experiment. Callers were also resident in the colony in the year prior to the experiment (except two individuals whose vocalizations were used in contact call playbacks).

## 2.5. Experimental trials

In total, we carried out 102 trials at 34 focal nest-boxes across the three study sites (15 nests at Stithians; 16 at Pencoose; three at Campus). Trials were conducted between 08.00 and 18.00. Trials at a single nest were carried out on consecutive days and at the same time of day (start times fell within 2 h for all three trials), to ensure a broadly similar rest period between trials for each nest, and control for variation in feeding rate over the course of the day. Trials at a focal nest-box began no earlier than 3 days after the first chick hatched (to minimize risk of nestling mortality or parental abandonment) and no later than 6 days post-hatching (to ensure provisioning was still frequent enough for parents to return within the trial period).

Prior to the experiment, the experimenter set up a chair and recording equipment 30 m away from the nest-box and directly facing the nest-box entrance. Recording equipment comprised a tripod with an HD camera (Panasonic HC-X920) taking a close-up view of the nest-box to identify colour rings of birds and record fine-scale behavioural measures at the nest. A small wide-angle camera (SJcam M10) was attached to the same tripod, to record experimenter behaviour and the behaviour of birds on approach to the nest-box. For the training phase, a loudspeaker was deployed on the ground halfway between the experimenter and the nest-box. After setting up equipment, the experimenter moved away to a concealed location to put on a mask and coat. The experimenter then returned to the area and approached the chair, keeping as far away from the nest-box as possible. The experimenter set camera recordings and a stopwatch timer for the end of the trial before sitting in the chair and remaining motionless for the duration of the trial, maintaining a constant gaze directly at the nest-box throughout. At the end of the trial the experimenter would get up, collect recording equipment and leave the area, keeping the mask on until out of sight of the colony.

Trials for the baseline (i) and test (iii) phase lasted for 30 min. In the training phase (ii), trials lasted until the first visit to the nest-box by any member of the focal pair (if birds did not return within 40 min, the trial was terminated). As the first bird made contact with the nest-box, the experimenter activated the playback using a handheld remote control. The experimenter would remain seated for 2 min after the playback and then leave the area, to ensure temporal consistency between presentation of the playback calls (unconditioned stimulus) and presentation of the human (conditioned stimulus) in the training phase.

If individuals hearing scold calls in the playback phase show a higher fear response in the test phase, it is possible that this effect could be carried over from having heard conspecific scold calls on the previous day. To control for this, all birds were exposed to the same number of scold and contact calls on the second day of the experiment using control playbacks. These were carried out by playing the other type of call from a speaker deployed in the same location, while birds were visiting the box but without human presence in the area.

## 2.6. Behavioural analysis

We recorded the frequency and duration of all behaviours exhibited at focal nest-boxes using the open-source video coding software BORIS [48]. Birds included in the experiment were individually identifiable. The behaviour of both males and females was recorded from experimental footage, with analysis carried out at the level of the individual (further details of video coding procedure, including inter-rater reliability, are given in the electronic supplementary material).

## 2.7. Statistical analysis

Of the 34 focal pairs, only birds that heard the playback were included in analyses. Playbacks occurred after the first bird landed on the nest-box in the second trial; this was the female of the pair in 24 cases and the male in 10 cases. However, in 13 cases the second member of the pair was close to the nest-box when the first bird triggered the playback. This meant that in total, 18 males and 29 females heard the playback and could potentially respond to the experimental treatments.

There was substantial heterogeneity of variance between males and females for many of the behaviours recorded. Females were more variable than males in their latency to return to the nest on the first visit (females: median = 26.4 s, quartile 1 (Q1) = 13.0 s, Q3 = 118.8 s; males: median = 30.6 s, Q1 = 7.9 s, Q3 = 48.3 s; electronic supplementary material, figure S3). Females also spent more time in the box over the whole trial than males (females: median = 527.4 s, Q1 = 69.9 s, Q3 = 983.4 s; males: median = 27.7 s, Q1 = 0.95 s, Q3 = 62.8 s; electronic supplementary material, figure S3), as would be expected given that females invest heavily in incubation at this stage in the breeding attempt. For these reasons, only data from females were analysed; for males, the smaller sample size and uneven distribution between treatment groups precluded formal analysis.

All analyses were carried out in R [49] using general linear mixed models (GLMMs) with a Gaussian error distribution, following box-cox transformation of response variables. All models included phase (baseline/test), call type (contact call/scold call) and caller familiarity (familiar/unfamiliar) as fixed effects with a three-way interaction, and individual identity as a random term. Models were constructed using the lme4 [50] package. Model plots were examined to ensure that assumptions were met (homogeneity and normality of residuals) and model goodness-of-fit estimates (marginal and conditional $R^2$) [51] were calculated using MuMIn [52]. Models were compared using log-likelihood ratio tests. Sample sizes vary between models (see below), after excluding birds that did not hear the playback in phase (ii), birds that did not exhibit the behaviour during the baseline (i) or test (iii) trials, and cases where reliable measures could not be obtained (see the electronic supplementary material).

### 2.7.1. Time taken to approach nest-box

In a similar experiment, Davidson *et al.* [24] found that jackdaws return to their nest-box more quickly after learning that a person is dangerous, which is likely to reflect a motivation to defend the nest and monitor the threat. To investigate how quickly females approach the nest-box when watched by a person previously associated with scold versus contact calls, we calculated the time females spent near the box prior to landing on the box on their first visit. This model included data from 21 females (10 females in the contact call group; 11 in the scold call group; see the electronic supplementary material).

### 2.7.2. Latency to enter nest-box

To determine whether females spend more time on the nest-box prior to entering when faced with a 'dangerous' person, we analysed female latency to enter the nest-box after landing. This model included data from 23 females (12 in the contact call group, 11 in the scold call group; see the electronic supplementary material).

### 2.7.3. Time spent in nest-box

Finally, we investigated whether females spent less time in the nest-box incubating and feeding chicks during presentations of a 'dangerous' person. This model included the total time spent in the nest-box by females in each trial as the response variable, using data from 20 females (12 in the contact call group and eight in the scold call group; see the electronic supplementary material).

## 3. Results

During the training phase (ii), birds responded to playbacks as would be expected for genuine scold calls and contact calls. Upon hearing scold calls, birds always left the nest-box immediately and several emitted scold calls in response. By contrast, birds did not leave the nest-box in response to contact calls, instead remaining on the nest-box during the playback or entering the nest-box to feed chicks.

### 3.1. Time taken to approach nest-box

Between the baseline phase (i) and the test phase (iii), females altered the time taken to approach their nest-box depending on the type of calls heard in the playback phase (ii) (phase/playback type interaction: $\chi^2_1 = 4.35$, $p = 0.037$; table 1). Females were quicker to approach the nest-box in the test phase compared to the baseline phase if they heard scold calls in the playback phase; females that heard contact calls took longer to approach the box in the test compared to their baseline (figure 2, table 1). For subjects in the scold call treatment, this equates to a 53% reduction in approach time on average between the baseline and the test phase, whereas subjects in the contact call treatment increased their approach time by 63% on average (electronic supplementary material, figure S4). Familiarity with the playback caller had no significant effect on the time taken to approach the nest-box ($\chi^2_1 = 0.68$, $p = 0.411$; table 1).

### 3.2. Latency to enter nest-box

The time taken by females to enter the nest-box after first landing did not change significantly between the baseline and test phase ($\chi^2_1 = 0.40$, $p = 0.53$). The type of call heard in the playback (scold calls/contact calls) and familiarity with the caller (familiar/unfamiliar) also had no significant effect on entry latency (playback type: $\chi^2_1 = 0.54$, $p = 0.46$; familiarity: $\chi^2_1 = 1.42$, $p = 0.23$; table 1). In the baseline phase (i), females in the scold call group were quicker to enter the nest-box than those in the contact call group (median entry latency ± inter-quartile range: scold call group = 6.5 s ± 50.25; contact call group = 21.8 s ± 430.68), despite chicks being of the same age in both treatment groups (see supplementary data [53]). However, female entry latency showed no significant change between trials as a result of the playback (phase/call type interaction: $\chi^2_1 = 0.60$, $p = 0.44$; table 1; electronic supplementary material, figure S5).

### 3.3. Time spent in nest-box

Females spent a similar length of time inside the nest-box during the baseline and test phase ($\chi^2_1 = 0.02$, $p = 0.89$). This was not influenced by the type of call heard in the playback ($\chi^2_1 = 1.20$, $p = 0.27$) or familiarity with the caller ($\chi^2_1 = 0.01$, $p = 0.94$; table 1; electronic supplementary material, figure S6).

### 3.4. Frequency of scolding

There were eight trials where females responded by scolding the experimenter (electronic supplementary material, figure S7). Five females scolded in the baseline phase (trial (i)) and three females scolded in the test phase (trial (iii)). One female scolded in both the baseline and test phases; four birds scolded in the baseline phase but not in the test phase, and two birds scolded in the test phase but not in the baseline phase (one after hearing contact calls and one after hearing scold calls in the playback phase). Females that scolded gave a median of 5.5 calls (range 2–11 calls per bird).

## 4. Discussion

We predicted that if jackdaws use social learning to inform their response to unfamiliar people, subjects would show a heightened response to a human watching their nest-box after seeing that person

**8**

**Table 1.** Model output for GLMMs investigating (*a*) the time taken by females to approach the nest-box on their first visit; (*b*) the latency for females to enter the nest-box after first landing; and (*c*) the time spent by females inside the nest-box over the whole trial. (Models investigated the change in response between trials (i)/(iii)) according to playback type (scold calls/contact calls) and caller familiarity (familiar/ unfamiliar). Values are taken from full models, with significant effects given in italics. Marginal $R^2$ ($R^2_m$) estimates the proportion of variance explained by the fixed effects, and conditional $R^2$ ($R^2_c$) estimates the proportion of variance explained by both the fixed and random effects.)

| model | n | subjects | $R^2_m$ | $R^2_c$ | fixed and random effects | $\beta \pm$ s.e. | t-value | variance ± s.d. |
|---|---|---|---|---|---|---|---|---|
| (*a*) approach time | 42 | 21 | 0.29 | 0.55 | intercept | 4.14 ± 0.45 | 9.23 | |
| | | | | | *phase* | *−0.13 ± 0.51* | *−0.26* | |
| | | | | | *playback type* | *0.33 ± 0.69* | *0.48* | |
| | | | | | familiarity | −0.47 ± 1.00 | −0.47 | |
| | | | | | *phase × playback type* | *−0.85 ± 0.77* | *−1.10* | |
| | | | | | phase × familiarity | 2.37 ± 1.13 | 2.09 | |
| | | | | | playback type × familiarity | −0.44 ± 1.26 | −0.35 | |
| | | | | | phase × playback type × familiarity | −2.79 ± 1.43 | −1.96 | |
| | | | | | subject | | | 0.58 ± 0.76 |
| | | | | | residual | | | 1.02 ± 1.01 |
| (*b*) entry latency | 46 | 23 | 0.10 | 0.61 | intercept | 2.60 ± 0.55 | 4.75 | |
| | | | | | phase | −0.65 ± 0.51 | −1.28 | |
| | | | | | playback type | −0.58 ± 0.87 | −0.67 | |
| | | | | | familiarity | 0.56 ± 1.10 | 0.51 | |
| | | | | | phase × playback type | 0.09 ± 0.80 | 0.11 | |
| | | | | | phase × familiarity | 0.74 ± 1.02 | 0.73 | |
| | | | | | playback type × familiarity | −0.57 ± 1.48 | −0.38 | |
| | | | | | phase × playback type × familiarity | 0.62 ± 1.37 | 0.45 | |
| | | | | | subject | | | 1.54 ± 1.24 |
| | | | | | residual | | | 1.16 ± 1.08 |

(*Continued.*)

**Table 1.** (*Continued.*)

| model | n | subjects | $R^2_m$ | $R^2_c$ | fixed and random effects | $\beta \pm$ s.e. | t-value | variance $\pm$ s.d. |
|---|---|---|---|---|---|---|---|---|
| (c) time in box | 40 | 20 | 0.10 | 0.66 | intercept | 16.45 ± 3.87 | 4.25 | |
| | | | | | phase | 2.85 ± 3.35 | 0.85 | |
| | | | | | playback type | 6.81 ± 6.00 | 1.13 | |
| | | | | | familiarity | −1.47 ± 6.00 | −0.25 | |
| | | | | | phase × playback type | −9.03 ± 5.19 | −1.74 | |
| | | | | | phase × familiarity | −1.30 ± 5.20 | −0.25 | |
| | | | | | playback type × familiarity | 3.01 ± 10.00 | 0.31 | |
| | | | | | phase × playback type × familiarity | 6.59 ± 8.30 | 0.80 | |
| | | | | | subject | | | 65.79 ± 8.11 |
| | | | | | residual | | | 39.29 ± 6.27 |

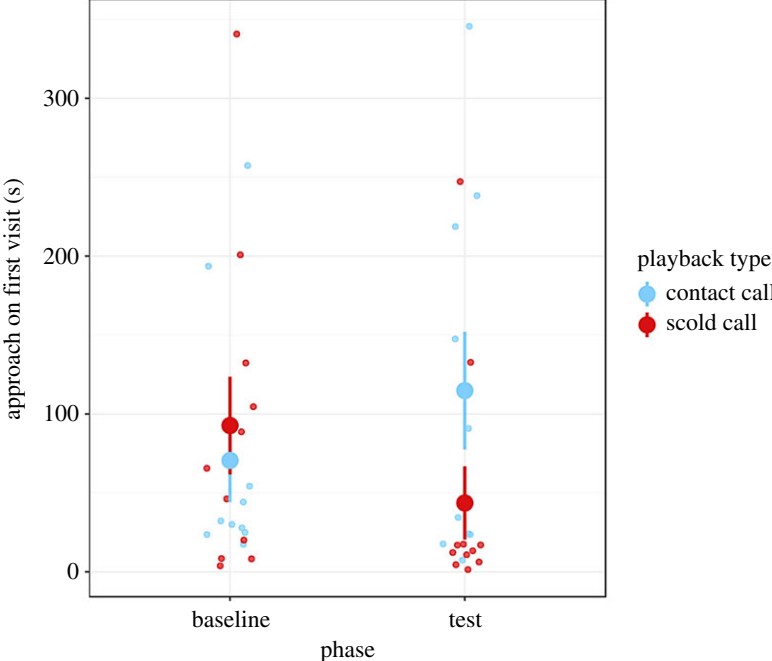

**Figure 2.** Time taken by females to land on the nest-box following their return to the area. Females that heard scold call playbacks in trial (ii) (training phase) were quicker to approach the nest in the test phase (trial (iii)) compared to their baseline (trial (i)). Females that heard contact call playbacks in the training phase took longer to approach the nest on their first visit in the test phase (trial (iii)) compared to their baseline (trial (i)). Points and whiskers denote mean and standard error, $n = 42$ observations of 21 females.

associated with conspecific scold calls. We found evidence to support this hypothesis, as females that heard scold call playbacks in the training phase (trial (ii)) spent less time in the area before landing on the nest-box in the test phase (trial (iii)) compared to the baseline phase (trial (i)). By contrast, females that heard contact call playbacks during the training phase (ii) spent longer near their nest-box before landing in the test phase (iii) compared to their baseline (i). Playback treatment (contact calls/scold calls) did not influence the latency of females to enter the nest-box after first landing, or the time spent in the nest-box overall.

The finding that subjects in the scold call group returned to the nest more quickly than subjects in the contact call group may reflect parents' motivation to monitor the perceived threat and defend the nest-box, and concurs with findings of similar studies in this species [24]. Returning to the nest may also allow individuals to gather more information before deciding to recruit others to mob the potential 'predator'. Indeed, when field researchers at our study sites visit nests to monitor breeding progress and/or weigh chicks, adult jackdaws commonly respond by returning to the area to monitor and scold the person (V. E. Lee and G. E. McIvor 2016-2017, personal observation). If jackdaws preferentially attend to social information from familiar individuals, we predicted that changes in response between the baseline (i) and test (iii) phases would be highest for birds that were trained using familiar scold calls (ii). We found no evidence to support this, as familiarity with the caller had no significant effect on any of the behavioural responses measured. This may be because for breeding birds hearing scold calls near the nest-box is a highly salient stimulus and thus always elicits a strong response. This explanation is supported by the fact that during the playback phase, all birds responded to the scold call playbacks immediately by leaving the nest-box and/or giving responsive scolds, whereas subjects that heard contact calls remained on the nest-box during the playback. In contrast to the scold call group, females in the contact call group showed an increase in latency to approach the nest between the baseline (i) and test (iii) phase. This may be because, in the absence of information that the person near the nest-box is dangerous, birds show reduced motivation to return quickly to the nest as the chicks grow and can be left alone for longer periods of time.

Individual test subjects also varied in their behavioural responses during experimental trials, with individual identity explaining a substantial proportion of the variance in the data (table 1). This may partly explain the modest overall effect size seen for the interaction between phase (baseline/test) and playback type (scold calls/contact calls) on the time taken by females to approach the nest-box (table 1). Taken together, our findings suggest that there is an element of social learning involved in

refining jackdaws' responses to unfamiliar people, but there is a considerable amount of individual variation in how these responses are manifested (see also [46]). Therefore, although jackdaws appear to use social learning to identify a 'dangerous' person, individuals vary in how they respond during subsequent encounters with that person. Individual variation in behaviour is often controlled for but rarely discussed explicitly in experiments of this kind, and is likely to be critical to our understanding of animal cognition [54,55].

This study provides direct evidence that individual animals alter their responses to individual people via social learning. Our findings complement those of Davidson *et al.* [24] who demonstrated that jackdaws' personal experience with individual people informs their subsequent behavioural response in a similar way. The current study extends this work, suggesting that jackdaws may not need to directly experience an unpleasant event to identify a human as 'dangerous' and can use social cues to learn about dangers associated with specific people. Our results also contribute to a wider body of work on socially acquired predator avoidance. Although this area has received relatively little attention compared to other aspects of social learning [2], previous studies have shown that social cues play an important role in learning to avoid novel predators [6,56,57], novel parasites [58,59] and dangerous locations [8]. To date, only one other study has investigated socially acquired predator avoidance in the wild, providing compelling circumstantial evidence for social transmission of information about dangerous people from informed to naive American crows [12]. Our experiment builds on this by showing that a single short-lived, commonly-occurring alarm calling event may be sufficient to alter individual behaviour in response to specific people, by reducing latency to return to the nest. These types of social learning events are likely to be important for organisms that exploit human-dominated habitats where individual people represent varying levels of threat, especially for species such as corvids that are often persecuted as pests.

Understanding how social learning shapes antipredator responses is vital to predicting and mitigating the effects of human activity, and provides valuable insights into how cognitive abilities influence adaptation to changing environments. However, socially acquired predator avoidance has received surprisingly little attention given its importance for individual fitness. Despite being limited in number, studies to date have provided some compelling insights into how animals learn socially about danger; further research is urgently needed to investigate how social environments facilitate antipredator learning, particularly under natural conditions. Studying a wide range of species is also essential to identify factors underlying success in anthropogenic habitats, which could subsequently be applied to effectively manage pest species and conserve declining populations.

Ethics. This work was carried out with approval of the University of Exeter research ethics committee (2017/1680) and adhered to the Association for the Study of Animal Behaviour Guidelines for the Use of Animals in Research. No birds were handled as part of this study, but birds were previously handled and marked under licence from the British Trust for Ornithology (C6449, C5752 and C6079) and the UK Home Office (project licence 30/3261).

Data accessibility. Data and R scripts associated with this work are available via Figshare (doi:10.6084/m9.figshare. 7828781) [53].

Authors' contributions. V.E.L. and A.T. designed the experiment; V.E.L. and N.R. ran the experimental trials; V.E.L. analysed the data; G.E.M. maintained field sites/study populations; V.E.L. and A.T. wrote the manuscript, with input from G.E.M.

Competing interests. We declare we have no competing interests.

Funding. V.E.L was supported by the Natural Environment Research Council (no. NE/L002434/1); N.R. was supported by Explo'ra Sup and AMI scholarships awarded by La Région Auvergne-Rhône-Alpes; A.T. and G.E.M. were supported by a BBSRC David Phillips Fellowship awarded to A.T. (no. BB/H021817/1; no. BB/H021817/2).

Acknowledgements. We are most grateful to the Gluyas family and staff at Pencoose Farm, Odette Eddy and the people of Stithians where the experiment was carried out. We would also like to thank Gabrielle Davidson and Andy Radford for valuable discussions on the experimental design. Richard Woods, Charlie Savill, Jenny Coomes and several student volunteers helped to collect call recordings, Beki Hooper acted as an independent coder for video analysis, and Alison Greggor assisted with ringing the birds involved in this study.

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
