## [Reviewer comments · Royal Society Open Science]

Review History

RSOS-191031.R0 (Original submission)

Review form: Reviewer 1 (Thomas Bugnyar)

Is the manuscript scientifically sound in its present form?

Yes

Are the interpretations and conclusions justified by the results?

Yes

Is the language acceptable?

Yes

Do you have any ethical concerns with this paper?

No

Have you any concerns about statistical analyses in this paper?

No

Recommendation?

Accept with minor revision (please list in comments)

Comments to the Author(s)

Lee et al. present an important study on an interesting topic, social learning about dangerous people. They show that wild jackdaws pick up on the information provided in played back calls of conspecifics that were given in the presence of an unfamiliar human experimenter and subsequently treat this human either as a threat (test) or not (control). The study nicely adds to the findings of horizontal and vertical information transmission about dangerous humans in American crows; yet, it is original because it tests the effect of scold calls under controlled experimental conditions.

The study is well designed and written up. The introduction follows a clear line of argumentation; the methods are explained thoroughly for most parts; the results are presented well and the discussion is balanced. Accordingly, I have only minor comments that should be easy to address in a revision.

p. 5, l. 108 Kern JM, Radford AN (2016) Biol. Lett. 12: 20160648 might be a relevant reference here.

p. 9, l. 175 Fig 1. I don't think this figure is needed; its legend is also partly redundant with the main text. I'd rather see Fig S2 (the masks) from the supplementary in the main text.

p. 11, l. 213-216: watch out, this might be understood wrongly (that you sometimes had scold calls of multiple individuals in one playback track). Try to phrase this part more clearly.

p. 12, 249: why did you present the playback from the ground?

p. 12, l. 251-252: how often was it not possible? Could proximity to the nest explain some of the remaining variation?

p. 13, l. 268-270: well done!

p. 14, 298-299: you included individual ID as a random term, does this mean random intercept? In this case, a random slope for individuals might be beneficial to the model design as well; some birds might stress easier than others.

p. 15, l. 303: maybe add '(see below)' after 'models'

p. 15, l. 307: this study could be explained a bit better in the intro already

p. 19, l. 356: any idea why there is this difference in the baseline phase? Age/need of chicks?

p. 20, l. 391: I wouldn't call it 'fear' response, as you didn't measure any emotions; why not just say 'response'?

p. 21, l. 408: again, delete 'fear'

p. 21, l. 427-428: maybe repeat here, what exactly you mean with individual variation in manifesting responses

p. 21, l. 431-432: I'd be careful with this statement. What do you mean with 'level of threat' (isn't it just threat/no threat)? I'd also tone down a bit the phrasing 'first direct experimental evidence', as the Marzluff studies had an experimental seeding component, too.

p. 22, l. 445: I agree, but maybe you'd elaborate on what these changes are specifically, i.e. return latencies rather than scolding; to me, that's the big difference to the Marzluff studies, where naïve crows started scolding as well – it might well be that to achieve this level, the birds would need (more often) exposure to (more) intensive scolding events, right?

Finally, a general remark: as far as I know, very similar setups were used on fish in the 90s (e.g. by Suboski and colleagues, just without a human as predator). Maybe you want to briefly refer to these in the discussion as well.

Review form: Reviewer 2

Is the manuscript scientifically sound in its present form?

Yes

Are the interpretations and conclusions justified by the results?

Yes

Is the language acceptable?

Yes

Do you have any ethical concerns with this paper?

No

Have you any concerns about statistical analyses in this paper?

No

Recommendation?

Accept as is

Comments to the Author(s)

This study investigates whether scold calls facilitate the process in which individuals learn to recognize novel humans as a threat. Although social learning of predators has long been determined for birds, it has not been well tested for the recognition of different humans. The present results include some important data on this mechanism. As the manuscript has been well written, I only have a few concerns.

(1) The current experiments were conducted with playback of scold calls, which was used as a social cue for learning. However, to me, it was unclear whether jackdaws in natural situations usually help to defend other pairs' nests. Do you have any evidence/citation to support for this?

(2) Also, I want a more detailed description on their colonies. Was the membership of the colonies stable across years? All the individuals contributed to the playback stimuli were alive during the experiments? As the authors used the calls recorded in 1-3 years before the experiments, it is important to describe these things to ensure the validity of experimental design.

(3) This paper mainly focusses on associated learning between humans and risks, but it would be worth to cite papers of social learning of cuckoos by their hosts (e.g., Davies & Welbergen, 2009).

Decision letter (RSOS-191031.R0)

02-Aug-2019

Dear Professor Lee

On behalf of the Editors, I am pleased to inform you that your Manuscript RSOS-191031 entitled "Social learning about dangerous people by wild jackdaws" has been accepted for publication in Royal Society Open Science subject to minor revision in accordance with the referee suggestions. Please find the referees' comments at the end of this email.

The reviewers and handling editors have recommended publication, but also suggest some minor revisions to your manuscript. Therefore, I invite you to respond to the comments and revise your manuscript.

- Ethics statement

- Data accessibility

<http://datadryad.org/submit?journalID=RSOS&manu=RSOS-191031>

- Competing interests

- Authors' contributions

- Acknowledgements

- Funding statement

Please ensure you have prepared your revision in accordance with the guidance at <https://royalsociety.org/journals/authors/author-guidelines/> -- please note that we cannot publish your manuscript without the end statements. We have included a screenshot example of

the end statements for reference. If you feel that a given heading is not relevant to your paper, please nevertheless include the heading and explicitly state that it is not relevant to your work.

Because the schedule for publication is very tight, it is a condition of publication that you submit the revised version of your manuscript before 11-Aug-2019. Please note that the revision deadline will expire at 00.00am on this date. If you do not think you will be able to meet this date please let me know immediately.

Please note that Royal Society Open Science charge article processing charges for all new submissions that are accepted for publication. Charges will also apply to papers transferred to Royal Society Open Science from other Royal Society Publishing journals, as well as papers

submitted as part of our collaboration with the Royal Society of Chemistry (<http://rsos.royalsocietypublishing.org/chemistry>).

If your manuscript is newly submitted and subsequently accepted for publication, you will be asked to pay the article processing charge, unless you request a waiver and this is approved by Royal Society Publishing. You can find out more about the charges at <http://rsos.royalsocietypublishing.org/page/charges>. Should you have any queries, please contact opencience@royalsociety.org.

Kind regards,
Alice Power
Editorial Coordinator
Royal Society Open Science
opencience@royalsociety.org

on behalf of Dr Alecia Carter (Associate Editor) and Kevin Padian (Subject Editor)
opencience@royalsociety.org

Associate Editor Comments to Author (Dr Alecia Carter):

Decision on RSOS-191031

I have now received two reviews of your manuscript and have read it myself. We all found your study interesting, well-executed and well-written. In particular, I found the study design to be very thorough, and very well thought-through. The reviewers have provided some constructive feedback and I also have some very minor comments to add that you will find below.

Minor comments:

L88: why “cultural transmission” here? If one is to use this term, it should be defined and explained and provide a reference.

L117: which -> that

LL144, 227, 406: whose pers. obs.?

L245: remove “of”

L275: briefly describe “BORIS” e.g. a programme for collecting behavioural data

LL288, 290: Better to present Q1 and Q3 rather than the range of the IQR as it is not symmetrical.

L350: Are these the back-transformed data (a box-cox transformation was done, L296)? It looks like this is not the case, so please present the median and IQR rather than the mean and SE (L350) as this is more appropriate for skewed data.

L357: that -> than

L384: 5 -> Five (digits should not start a sentence)

Reviewer comments to Author:

Reviewer: 1

Lee et al. present an important study on an interesting topic, social learning about dangerous people. They show that wild jackdaws pick up on the information provided in played back calls of conspecifics that were given in the presence of an unfamiliar human experimenter and

subsequently treat this human either as a threat (test) or not (control). The study nicely adds to the findings of horizontal and vertical information transmission about dangerous humans in American crows; yet, it is original because it tests the effect of scold calls under controlled experimental conditions.

The study is well designed and written up. The introduction follows a clear line of argumentation; the methods are explained thoroughly for most parts; the results are presented well and the discussion is balanced. Accordingly, I have only minor comments that should be easy to address in a revision.

p. 5, l. 108 Kern JM, Radford AN (2016) *Biol. Lett.* 12: 20160648 might be a relevant reference here.

p. 9, l. 175 Fig 1. I don't think this figure is needed; its legend is also partly redundant with the main text. I'd rather see Fig S2 (the masks) from the supplementary in the main text.

p. 11, l. 213-216: watch out, this might be understood wrongly (that you sometimes had scold calls of multiple individuals in one playback track). Try to phrase this part more clearly.

p. 12, 249: why did you present the playback from the ground?

p. 12, l. 251-252: how often was it not possible? Could proximity to the nest explain some of the remaining variation?

p. 13, l. 268-270: well done!

p. 14, 298-299: you included individual ID as a random term, does this mean random intercept? In this case, a random slope for individuals might be beneficial to the model design as well; some birds might stress easier than others.

p. 15, l. 303: maybe add '(see below)' after 'models'

p. 15, l. 307: this study could be explained a bit better in the intro already

p. 19, l. 356: any idea why there is this difference in the baseline phase? Age/need of chicks?

p. 20, l. 391: I wouldn't call it 'fear' response, as you didn't measure any emotions; why not just say 'response'?

p. 21, l. 408: again, delete 'fear'

p. 21, l. 427-428: maybe repeat here, what exactly you mean with individual variation in manifesting responses

p. 21, l. 431-432: I'd be careful with this statement. What do you mean with 'level of threat' (isn't it just threat/no threat)? I'd also tone down a bit the phrasing 'first direct experimental evidence', as the Marzluff studies had an experimental seeding component, too.

p. 22, l. 445: I agree, but maybe you'd elaborate on what these changes are specifically, i.e. return latencies rather than scolding; to me, that's the big difference to the Marzluff studies, where naïve crows started scolding as well – it might well be that to achieve this level, the birds would need (more often) exposure to (more) intensive scolding events, right?

Finally, a general remark: as far as I know, very similar setups were used on fish in the 90s (e.g. by Suboski and colleagues, just without a human as predator). Maybe you want to briefly refer to these in the discussion as well.

Reviewer: 2

Comments to the Author(s)

This study investigates whether scold calls facilitate the process in which individuals learn to recognize novel humans as a threat. Although social learning of predators has long been determined for birds, it has not been well tested for the recognition of different humans. The present results include some important data on this mechanism. As the manuscript has been well written, I only have a few concerns.

(1) The current experiments were conducted with playback of scold calls, which was used as a social cue for learning. However, to me, it was unclear whether jackdaws in natural situations usually help to defend other pairs' nests. Do you have any evidence/citation to support for this?

(2) Also, I want a more detailed description on their colonies. Was the membership of the colonies stable across years? All the individuals contributed to the playback stimuli were alive during the experiments? As the authors used the calls recorded in 1-3 years before the experiments, it is important to describe these things to ensure the validity of experimental design.

(3) This paper mainly focusses on associated learning between humans and risks, but it would be worth to cite papers of social learning of cuckoos by their hosts (e.g., Davies & Welbergen, 2009).

Author's Response to Decision Letter for (RSOS-191031.R0)

See Appendix A.

Decision letter (RSOS-191031.R1)

12-Aug-2019

Dear Professor Lee,

I am pleased to inform you that your manuscript entitled "Social learning about dangerous people by wild jackdaws" is now accepted for publication in Royal Society Open Science.

on behalf of Dr Alecia Carter (Associate Editor) and Kevin Padian (Subject Editor)
openscience@royalsociety.org

Associate Editor Comments to Author (Dr Alecia Carter):

I'm happy for Figure 1 to remain in the main text (I agree it makes the design and predictions very clear). I'm also happy for Supplementary Figure 2 to appear in the main article.

Appendix A

RSOS-191031: Lee, V. E., Régli, N., Mclvor, G. E., & Thornton, A. Social learning about dangerous people by wild jackdaws

Response to referee comments

Response to Associate Editor Comments

I have now received two reviews of your manuscript and have read it myself. We all found your study interesting, well-executed and well-written. In particular, I found the study design to be very thorough, and very well thought-through. The reviewers have provided some constructive feedback and I also have some very minor comments to add that you will find below.

We are glad that you found our study interesting and thank you for your comments, which we have incorporated into the manuscript as detailed below.

Minor comments:

L88: why “cultural transmission” here? If one is to use this term, it should be defined and explained and provide a reference.

For clarity and brevity, this line now reads “...horizontal and vertical transmission of social information” (L88).

L117: which -> that

This change has been made (L116).

LL144, 227, 406: whose pers. obs.?

Initials of the authors have been added (V.E.L. and G.E.M., L146, L230, L413). Both have several years of experience working with these populations, especially G.E.M. who has been observing these birds over the lifetime of the project.

L245: remove “of”

Done (L250).

L275: briefly describe “BORIS” e.g. a programme for collecting behavioural data

This line now reads “...using the open-source video coding software BORIS” (L281).

LL288, 290: Better to present Q1 and Q3 rather than the range of the IQR as it is not symmetrical.

Done (L295, L297).

L350: Are these the back-transformed data (a box-cox transformation was done, L296)? It looks like this is not the case, so please present the median and IQR rather than the mean and SE (L350) as this is more appropriate for skewed data.

Done (L364).

L357: that -> than

Done (L363).

L384: 5 -> Five (digits should not start a sentence)

Done (L389).

Response to Reviewer #1

Lee et al. present an important study on an interesting topic, social learning about dangerous people. They show that wild jackdaws pick up on the information provided in played back calls of conspecifics that were given in the presence of an unfamiliar human experimenter and subsequently treat this human either as a threat (test) or not (control). The study nicely adds to the findings of horizontal and vertical information transmission about dangerous humans in American crows; yet, it is original because it tests the effect of scold calls under controlled experimental conditions.

The study is well designed and written up. The introduction follows a clear line of argumentation; the methods are explained thoroughly for most parts; the results are presented well and the discussion is balanced. Accordingly, I have only minor comments that should be easy to address in a revision.

We are glad that you find our study interesting, and thank you for your helpful and constructive feedback. We have incorporated the changes that you suggest into the manuscript, and we address individual comments below.

p. 5, l. 108 Kern JM, Radford AN (2016) Biol. Lett. 12: 20160648 might be a relevant reference here.

Thank you for pointing this out, we have incorporated this citation into L107.

p. 9, l. 175 Fig 1. I don't think this figure is needed; its legend is also partly redundant with the main text. I'd rather see Fig S2 (the masks) from the supplementary in the main text.

We would prefer Figure 1 to appear in the main text to clarify the experimental design, but would be willing to move Figure S2 to the main body of the manuscript in addition to/instead of Figure 1. We defer to the Editor's preference on this point.

p. 11, l. 213-216: watch out, this might be understood wrongly (that you sometimes had scold calls of multiple individuals in one playback track). Try to phrase this part more clearly.

Thank you for pointing this out. We have clarified this line: "Extracted calls were arranged into playback files comprising either 5 contact calls or 5 scold calls from a single individual, occurring at 2s intervals to simulate natural calling (Supplementary Figure S1). Where possible, 5 unique calls were used in playbacks" (L213).

p. 12, 249: why did you present the playback from the ground?

Whilst we would have preferred to present playbacks from trees, variation between nests meant that the ground was the only playback location that could be standardised across all nests. This was also done to ensure that playbacks were presented from the direction of the experimenter, thereby

directing subjects' attention towards the person and strengthening the association between the social information and the experimenter's presence. As we mention in L333, all subjects responded to playbacks as they would to genuine contact/scold calls, suggesting that presenting the playback from the ground did not diminish the ecological relevance of the stimulus.

p. 12, l. 251-252: how often was it not possible? Could proximity to the nest explain some of the remaining variation?

We realise that the wording in the manuscript may have been slightly misleading. Experimenters never walked directly underneath the nest box, but because there were a limited number of directions from which a given nest box could be approached, experimenters passed by some nest boxes more closely than others when setting up the experiment. Steps were taken to ensure that nest boxes were not approached more closely than was strictly necessary (no closer than 5m), and that the same route was taken for all trials at a given nest box. Moreover, as nest boxes that were approached more closely during setup also experience higher human disturbance in general (our study sites are located on working farmland), individuals may differ in their tolerance of disturbance in the first instance. We designed this repeat-measures experiment to control for these differences by investigating individual change in behaviour over time: whilst we recognise that these responses may be influenced by experimenter proximity we are unable to quantify this using our existing data, and it is likely that a number of other factors (e.g. individual differences in tolerance to disturbance) also contribute to the remaining variation in the data.

We have clarified the main text, which now reads: "The experimenter then returned to the area and approached the chair, keeping as far away from the nest box as possible" (L257).

p. 13, l. 268-270: well done!

Thank you!

p. 14, 298-299: you included individual ID as a random term, does this mean random intercept? In this case, a random slope for individuals might be beneficial to the model design as well; some birds might stress easier than others.

We agree with the reviewer on this point, and were interested to learn whether responses to the experiment differed between individuals as well as between treatments. During analysis, we initially included random slopes but were unable to fit models adequately due to lack of power.

p. 15, l. 303: maybe add '(see below)' after 'models'

Done, this line now reads: "Sample sizes vary between models (see below)..." (L309).

p. 15, l. 307: this study could be explained a bit better in the intro already

We have elaborated on this study in the introduction as suggested. L120 now reads: "When presented with a person wearing a mask, jackdaws will return to their nest more quickly if that mask had previously been worn during a nest intrusion, demonstrating that jackdaws learn to recognise individual people and associate them with prior events [24]."

p. 19, l. 356: any idea why there is this difference in the baseline phase? Age/need of chicks?

It had also occurred to us that this effect may be influenced by chick age, e.g. females in the scold call group may be entering the nest box more quickly if their chicks are younger and more vulnerable than those in the contact call group. However, there are no differences in mean or median chick age between treatment groups in the baseline phase.

For interested readers, we have updated the R script to illustrate this point (code line 470-476), and have clarified the main text which now reads: “In the baseline phase (1), females in the scold call group were quicker to enter the box than those in the contact call group (median entry latency±IQR: scold call group=6.5s±50.25; contact call group=21.8s±430.68), despite chicks being of the same age in both treatment groups (see Supplementary data and scripts)” (L363).

With the data available, we are unable to ascertain the reason for the group differences in the baseline phase, which may simply reflect natural variation in female behaviour. Given that our repeat-measures design quantifies change in individual behaviour over time, this difference does not influence the results of our study.

p. 20, l. 391: I wouldn't call it 'fear' response, as you didn't measure any emotions; why not just say 'response'?

The word “fear” has now been removed from this sentence (L397).

p. 21, l. 408: again, delete 'fear'

Done (L414).

p. 21, l. 427-428: maybe repeat here, what exactly you mean with individual variation in manifesting responses

We have clarified this statement by including an extra sentence: “Therefore, although jackdaws appear to use social learning to identify a ‘dangerous’ person, individuals vary in how they respond during subsequent encounters with that person” (L435).

p. 21, l. 431-432: I'd be careful with this statement. What do you mean with 'level of threat' (isn't it just threat/no threat)? I'd also tone down a bit the phrasing 'first direct experimental evidence', as the Marzluff studies had an experimental seeding component, too.

This sentence has been changed, and now reads: “This study provides direct evidence that individual animals alter their responses to individual people via social learning” (L439).

p. 22, l. 445: I agree, but maybe you'd elaborate on what these changes are specifically, i.e. return latencies rather than scolding; to me, that's the big difference to the Marzluff studies, where naïve crows started scolding as well – it might well be that to achieve this level, the birds would need (more often) exposure to (more) intensive scolding events, right?

We agree with the reviewer on this point: the Cornell et al. (2012) study provides compelling evidence for social learning about dangerous people, but it may be that an extreme event (such as trapping crows) is required to elicit such a strong response (alarm calling and mobbing) via social learning alone. From our study, it appears that short-lived, commonly-occurring alarm calling events are sufficient for social learning to occur, and result in more subtle changes in individual behaviour.

We have highlighted the specific changes in L452: “Our experiment builds on this by showing that a single short-lived, commonly-occurring alarm calling event may be sufficient to alter individual behaviour in response to specific people, by reducing latency to return to the nest”.

Finally, a general remark: as far as I know, very similar setups were used on fish in the 90s (e.g. by Suboski and colleagues, just without a human as predator). Maybe you want to briefly refer to these in the discussion as well.

Thank you for bringing these studies to our attention, we have included the following study as a citation in L448:

*Suboski, M. D., Bain, S., Carty, A. E., McQuoid, L. M., Seelen, M. I., & Seifert, M. (1990). Alarm reaction in acquisition and social transmission of simulated-predator recognition by zebra danio fish (*Brachydanio rerio*). *Journal of Comparative Psychology*, 104(1), 101–112.*

Response to Reviewer #2

This study investigates whether scold calls facilitate the process in which individuals learn to recognize novel humans as a threat. Although social learning of predators has long been determined for birds, it has not been well tested for the recognition of different humans. The present results include some important data on this mechanism. As the manuscript has been well written, I only have a few concerns.

Thank you for your helpful review, we have addressed your concerns in the manuscript as detailed below.

(1) The current experiments were conducted with playback of scold calls, which was used as a social cue for learning. However, to me, it was unclear whether jackdaws in natural situations usually help to defend other pairs' nests. Do you have any evidence/citation to support for this?

To our knowledge, jackdaws do not cooperate to defend each other's nests. However, because jackdaws typically live at high densities (mentioned in L116) and respond to threats by mobbing, a potential threat identified anywhere in the vicinity of the colony will elicit scold calls from birds that can see this threat, and recruit birds to the area. For example, Woods et al. (2018) showed that playbacks of scold calls both in and outside nesting colonies will recruit birds from the colony to mob the 'predator'. Furthermore, the presence of researchers in our colonies often elicits scolding from birds nearby, and during the experimental trials the experimenter could be seen from multiple nest boxes as well as by other birds present in the area. Therefore, the occurrence of scold calls in association with a person near any nest box is likely to be an ecologically relevant stimulus for our test subjects, and does not require that birds cooperate to defend each other's nests.

(2) Also, I want a more detailed description on their colonies. Was the membership of the colonies stable across years? All the individuals contributed to the playback stimuli were alive during the experiments? As the authors used the calls recorded in 1-3 years before the experiments, it is important to describe these things to ensure the validity of experimental design.

The reviewer raises an excellent point. All callers were alive at the time of the experiment, and all had been resident in the colony for at least one year prior to the experiment being carried out (except two birds that provided contact calls, where familiarity was not expected to influence subjects' responses to the playback; these individuals were nesting in the colony at the time of the experiment, but whether they had resided in the colony in previous years is unknown).

To clarify this, we have added an extra sentence to the 'Allocation of playbacks' section of the Methods: "All callers were known to be alive and breeding in the colonies at the time of the experiment. Callers were also resident in the colony in the year prior to the experiment (except two individuals whose vocalisations were used in contact call playbacks)" (L235).

(3) This paper mainly focusses on associated learning between humans and risks, but it would be worth to cite papers of social learning of cuckoos by their hosts (e.g., Davies & Welbergen, 2009).

Thank you for pointing this out, we have cited these papers (Davies & Wellbergen, 2009; Feeney & Langmore, 2013) in L448.